# Measuring the Vaccine Success Index: A Framework for Long-Term Economic Evaluation and Monitoring in the Case of Rotavirus Vaccination

**DOI:** 10.3390/vaccines12111265

**Published:** 2024-11-08

**Authors:** Baudouin Standaert, Marc Raes, Olivier Ethgen, Bernd Benninghoff, Mondher Toumi

**Affiliations:** 1Department of Care & Ethics, Faculty of Medicine & Life Sciences, University of Hasselt, 3590 Diepenbeek, Belgium; 2Department of Immunology & Infection, Faculty of Medicine & Life Sciences, University of Hasselt, 3590 Diepenbeek, Belgium; fa303885@skynet.be; 3Department of Public Health Sciences, Faculty of Medicine, University of Liège, 4000 Liège, Belgium; o.ethgen@serfan.eu; 4A&N Immugen, 91349 Egloffstein, Germany; bernd.benninghoff@gmx.de; 5Public Health, University of Aix-Marseille, 13002 Marseille, France; mondher.toumi@emaud.eu

**Keywords:** rotavirus, vaccination, cost-effectiveness, cost-impact, success index

## Abstract

New vaccination programs measure economic success through cost-effectiveness analysis (CEA) based on an outcome evaluated over a certain time frame. The reimbursement price of the newly approved vaccine is then often reliant on a simulated ideal effect projection because of limited long-term data availability. This optimal cost-effectiveness result is later rarely adjusted to the observed effect measurements, barring instances of market competition-induced price erosion through the tender process. However, comprehensive and systematic monitoring of the vaccine effect (VE) for the evaluation of the real long-term economic success of vaccination is critical. It informs expectations about vaccine performance with success timelines for the investment. Here, an example is provided by a 15-year assessment of the rotavirus vaccination program in Belgium (RotaBIS study spanning 2005 to 2019 across 11 hospitals). The vaccination program started in late 2006 and yielded sub-optimal outcomes. Long-term VE surveillance data provided insights into the infection dynamics, disease progression, and vaccine performance. The presented analysis introduces novel conceptual frameworks and methodologies about the long-term economic success of vaccination programs. The CEA evaluates the initial target vaccination population, considering vaccine effectiveness compared with a historical unvaccinated group. Cost-impact analysis (CIA) covers a longer period and considers the whole vaccinated and unvaccinated population in which the vaccine has direct and indirect effects. The economic success index ratio of CIA over CEA outcomes evaluates long-term vaccination performance. Good performance is close to the optimal result, with an index value ≤1, combined with a low CEA. This measurement is a valuable aid for new vaccine introductions. It supports the establishment of robust monitoring protocols over time.

## 1. Introduction

The success of a vaccine is expressed as its vaccine effect (VE). This is referred to as ‘vaccine efficacy’ when assessed under the strict conditions of a double-blinded randomized controlled trial (RCT) evaluating vaccinated and unvaccinated subjects over a fixed period [1], and as ‘vaccine effectiveness’ when assessed under uncontrolled conditions comparing vaccinated and unvaccinated groups in real-world practice [2]. VE may change over time and may decrease, described as vaccine waning. Different types of waning can be identified [3]. In classical waning, the vaccine inoculum does not activate a sustainable immune response in the host, and this can be remedied by a vaccine booster dose. Another waning process is related to the vaccine’s biological activity, producing partial antigen activation, and resulting in lower VE over time after multiple exposures to the pathogen. Such leaky or partially responding effects have been described in studies analyzing influenza vaccines introduced in recent years [4,5,6]. A third type of waning, which differs from the previous two in that it is not caused by the vaccine, may appear with suboptimal introduction of a new vaccine. This type of waning, referred to in this paper as ‘strategic waning’, can only be detected by long-term measurement of VE [7], and it is the focus of this paper. This analysis proposes an approach to measuring the success of a vaccination program from an economic perspective across the whole population, including any strategic waning effect. Epidemiological assessments of infectious diseases and vaccinations are made at the level of comparative groups and use the term ‘effectiveness’, whereas evaluations at the level of the whole population use the term ‘impact’ assessment [8]. Health economic evaluations have not conventionally made this distinction; however, it could help to understand that vaccination may have a broader and longer economic impact on the whole population extending beyond specific comparable at-risk target groups [9,10]. Furthermore, a sub-optimal start to a vaccination program may affect the long-term impact of the vaccination program on the whole population. This paper illustrates this potential effect and highlights the importance of evaluating a vaccination program in a broader context than the conventional one. Most of the analyses and results presented here are based on rotavirus vaccine introduction in Belgium using data from the RotaBIS study. Many of the results have already been published elsewhere [7,9,11,12]. This paper builds on previous publications by making the link between effectiveness and impact and showing how the success or failure of a new vaccination program can be measured using a new ratio calculation, called the success index. This helps to assess the economic long-term value achieved with vaccination in complex environments. The paper uses cost-effectiveness and cost-impact analyses to obtain results that should help healthcare payers assess whether the investment in a vaccination program was successful [13].

## 2. Materials and Methods

### 2.1. Background

This analysis concerns rotavirus infection and its vaccination in a high-income country, Belgium. Rotavirus infection is distinctive in that it is seasonal in the northern hemisphere, spreading during each winter period from around January to March/April, and affects children up to 5 years old, causing severe diarrhea with a risk of dehydration [14]. Different sources of this infection exist, but the primary source is infants aged 3 to 14 months who spread the infection across the whole at-risk group. More exposure to infection increases the immunity level against the disease [15]. The infection is contagious, with a primary reproductive number of R_0_ estimated at around 9 under normal conditions [16,17]. The disease spread as a function of age has a clear Weibull distribution pattern, with a mean around 15 months of age and a long decreasing tail to the right up to 60 months of age [14]. Rotavirus infection has mainly been studied by the rate of cases leading to hospitalization, with less measurement of infection rates at the primary healthcare or family care level as laboratory tests are not systematically conducted. Two vaccines are available in high-income countries, a two-dose vaccine (Rotarix, GSK Biologicals, Wavre, Belgium) and a three-dose vaccine (Rotateq, Merck, Rahway, NJ, USA), which should be given at the age of 6 weeks for the first dose with a 4-week interval for subsequent doses [18,19,20]. As reported in clinical trials, vaccination provides a very high response rate in reducing severe cases [21,22]. However, the vaccination should be given prior to 32 weeks of age as it may have a very low risk of the severe adverse effect of intussusception [23,24]. There is, therefore, no option to introduce a catch-up vaccine strategy to vaccinate everyone at risk at once. Consequently, every newborn needs to be vaccinated, and the time required to cover the whole at-risk group (children up to 5 years old) with vaccination may be at least 5 years. The possibility of vaccine waning over time has been suggested, based on comparing the first- and second-year efficacy data [25]. When introduced in a child population in the defined age group (aged 6 weeks to 8 months), the vaccine diminishes infection in the primary source (infants aged 3–14 months). Consequently it causes an indirect protective effect amongst the unvaccinated individuals in the population. This also reduces the potential for creating secondary sources of infection in the same target group, i.e., infection sources developed as a consequence of the primary source. So the primary source is not only the cause of direct infection but also the cause of secondary infection [11]. If the vaccine substantially reduces the primary source of infection by achieving a high coverage rate at start of the vaccination program, it can also reduce the secondary sources of infection, thus causing an important additional indirect effect. If the vaccine coverage is lower and/or the timing of the program’s introduction is sub-optimal in relation to the disease’s seasonality, it allows a greater manifestation of secondary sources of infection that are not influenced by the vaccine’s direct effect. The level of secondary infection sources affects the vaccine’s effect over time and the endemicity level of rotavirus in the whole population. Increasing vaccine coverage has an immediate impact only on the initial primary source, with a lesser effect on secondary sources of infection that develop into new primary sources of infection in the population over time [7].

### 2.2. Economic Assessment

An economic value assessment of a vaccine, short to long term, must rely on two sources of information: data observations and credible modeling exercises.

#### 2.2.1. Data

The data observations are from the RotaBIS study, which started to assemble information in 2007, a year after the vaccine’s introduction. The vaccine price was 80% reimbursed by the Belgian authorities in November 2006 [26,27]. Hospital data on rotavirus infection in young children were retrospectively collected for the years of 2005 and 2006. The same information was further collected annually for 13 years (2007 to 2019) from 11 hospitals that were representative of the different parts of the country (9 were general hospitals with a pediatric ward, 2 were pediatric-only hospitals, and 4 of the 11 hospitals were academic centers). The data assembled for each event, in addition to the lab test results and dates of rotavirus detection, were the date of hospitalization, the age when the disease occurred, sex, duration of hospitalization, and nosocomial acquisition. The study protocol with the first data analyses was presented in 2011 [28]. The information relevant to the present analysis is summarized in Table 1, showing the numbers of disease-specific hospitalizations by age and year reported over a total period of 15 years (the pre-vaccination years of 2005 and 2006 are reported as average values for the two years combined).

#### 2.2.2. Modeling

Models must replicate the observed data (blue line in Figure 1A). They are constructed based on variables that affect the observed curve with direct and indirect vaccine effects [7]. To facilitate model construction, the observation period was split into two timeframes (period I and period II), with a different model type for each period (Figure 1B). Full details about the models, including sensitivity analyses, have been published elsewhere [7].

Vaccination period I is the vaccine uptake period, which may last 5 to 7 years until a new infection equilibrium has been reached in the target group of children aged ≤5 years. The model selected for that period uses a time-dependent regression equation that shapes the curve with the different forces that impact the regression line, reproducing the number of disease-specific hospitalizations observed per year. Two overall forces identify each combination of several components. The first overall force includes the direct vaccine effect, with components of effectiveness, coverage, and waning. The second overall force looks at the indirect effect of the vaccine, with components of herd effect and secondary sources of infection. The uptake period defined here is the measurement of vaccine ‘effectiveness’, which should be stable when the new infection equilibrium is reached in the target group at the end of period I.

Period I is subsequently followed by the post-uptake period (period II). During that second period, the dynamic infection spread is simulated through a time differential compartmental equation with susceptible, infectious, and recovered (SIR) groups. Those compartments are linked by transition rates replicating the observed biennial disease peaks over time using a Hamer model design [29]. The frequency and height of the peaks depend on the entry conditions defined at the end of period I. The entry conditions include the remaining infection rate in the population, the maintained vaccine coverage rate with its net effect, the susceptible group (newborns) entering at any given time point, and the contact matrix of the at-risk population. It is important to note that the initial primary source of infection pre-vaccination has now moved in the post-uptake period (period II) to an older age group. The shift is developed during the vaccine uptake period (period I) when the vaccine coverage and the timing of initiating the vaccination program are not optimal. The total period of period I + period II is defined here as the period over which the vaccine ‘impact’ assessment is calculated.

### 2.3. Cost-Effectiveness and Cost-Impact Analysis

The economic evaluations have an identical basic formula to assess the two measures of interest: the cost-effectiveness analysis (CEA) and the cost-impact analysis (CIA). CEA references the vaccine uptake period using the modeled or observed data as input until the post-uptake period is reached. CEA evaluates the initial target vaccination population, considering the effectiveness of the direct and indirect effects of the vaccine, such as the positive herd effect and the opposing effect of secondary sources of infection in unvaccinated individuals, and compares the results with a historical group that is unvaccinated [30]. CIA covers a more extended period than CEA, including the uptake and post-uptake periods, and the whole vaccinated and unvaccinated population in which the vaccine has direct and indirect effects (the impact of the beneficial herd effect will be less than in CEA and the impact of the detrimental secondary infection source effect will be greater) again over time. The accumulated results are compared with the situation before initiating the vaccination program (*pv*) [10]. The formula for *CIA* is as follows:$$CIA=\frac{∆C}{∆E}$$
$$∆C={(C}_{pv}-(C_{uv}+C_{v}))$$
$$∆E={(E}_{pv}-\left(E_{uv}+E_{v}\right))$$
where ∆ = difference; *C* = cost; *E* = health effect often expressed in quality-adjusted life-years (QALYs); *pv* = pre-vaccination; *uv* = unvaccinated; and *v* = vaccinated.

The negative indirect effect of vaccination on the whole population is added to the evaluation, as secondary sources of infection in older age groups develop into new primary sources of infection if the vaccination program’s initiation is not optimal. This evaluation method for *CIA* refers to impact assessment in epidemiology, as presented by Hanquet et al. [8], which is applied here as an economic assessment.

#### 2.3.1. Input Data

Table 2 presents the input data for the estimation of the cost and QALY loss due to rotavirus hospitalization. These data are based on Belgium. The cost data are those used when the vaccine was launched in 2006 and received its reimbursement price in Belgium, which has marginally changed over time.

Input values for the key variables that define the shape of the curve during the vaccine uptake period are presented in Table 3.

#### 2.3.2. Output Data

The output obtained is the incremental cost-effect ratio (ICER) achieved, with ‘effect’ defined as ‘effectiveness’ or ‘impact’ measurement for CEA or CIA, respectively. Scenario evaluations are performed comparing the ICER of the uptake period (7 years) with the ICIR obtained for the uptake + post-uptake period (15 years), using observed data. The ICIR obtained of the observed data is then compared with the ICIR of the optimal launch and the ICIR of the worst-case launch. Discounting cost or health gains were not applied because the present analysis is concerned with assessing the optimal strategy for vaccine launch and not with assessing the vaccine’s value.

The output for the optimal launch and the worst-case vaccine launch scenarios used the previous two models, where the vaccine coverage rates were adjusted in each scenario (86% from the start in the optimal launch scenario and 40% in the worst-case scenario) with a slightly lower VE (0.86 instead of 0.95). The lower VE was justified by the potential presence of the two vaccine waning processes over time (reduced immune response, leaky vaccine). Figure 2 illustrates the decrease in hospitalizations simulated with the model programs for an optimal vaccine launch and a worst-case scenario of a poor vaccination launch, together with the observed data.

### 2.4. The Success Index

The results of CEA and CIA do not reveal whether a vaccination program has been successful over time, with ‘success’ meaning that the vaccination is maintaining control of the infection spread, indicated by very low rates of disease-specific hospitalizations in the long term. Lower CEA results are often considered to have reached better outcomes. However, CEA is mainly performed to obtain a value assessment related to the acceptable vaccine price; therefore, a low CEA could also result from a low vaccine price and not from reasonable long-term control of infection spread achieved by the vaccination program. Reasonable control of infection spread would have the effect of reducing the frequency and height of the biennial peaks in the post-uptake period. There are, however, two situations in which those post-uptake peaks could be reduced. One results from optimal vaccine introduction with a high vaccine coverage at the start, inducing a high indirect effect with control of primary and secondary sources of infection, resulting in peaks of limited size and frequency in the long term. The other could result from deficient vaccine uptake, resulting in the continued dominance of the initial primary source of infection and consequently no clear manifestation of the secondary sources of infection in the long term. The two situations can be distinguished by considering the ratio of the results of CIA over CEA, here called the success index. A successful vaccination launch will have a ratio close to 1, with a low CEA result. A poor vaccination launch will also have a ratio close to 1, but with a high CEA result. Intermediate results indicate that the vaccination program was not a failure but could have been more successful with an optimal vaccine launch. The ratio will be close to 1.5, combined with a higher CEA result than that projected for the optimal launch. There is, therefore, a maximum ratio for CIA over CEA, dependent on the level of attenuation of the primary source of infection by the vaccination program.

## 3. Results

### 3.1. ICER Results

The cost-effectiveness results for the uptake period compared with no vaccination are shown in Table 4.

The results shown in Table 5 compare hospitalization days for the whole uptake period, including post-uptake data with results for no vaccination. The ICER result from this CIA differs from that calculated in the previous CEA (Table 4) due to the appearance of secondary hospitalization peaks at 9 and 11 years post-vaccine introduction, which negatively impacts VE over time.

### 3.2. The Success Index and the Scenario Analyses

Table 6 shows the success index ratio calculated from the CEA results (Table 4) and the CIA results (Table 5). Table 6 also presents the simulation results for the two scenarios of optimal vaccine launch and worst-case vaccine launch. Figure 3 shows the simulation results for the success index ratio expressed as a function of the CEA value calculated over a pre-specified evaluation period (defined in this analysis as 15 years). The results follow a lognormal outline distribution, illustrating the success and failure areas for this setting when the threshold for the CIA/CEA ratio has been set at 1 (CEA_max_ = €15,091 for S; CEA_min_ = €60,296 for F). The results for the optimal launch scenario, the worst-case launch scenario, and the observed Belgian data are plotted as point values; the optimal launch scenario falls into the success ‘S’ area of the outline, the worst-case launch scenario falls into the failure ‘F’ area, and the observed data from Belgium fall into an intermediate area (Figure 3).

## 4. Discussion

The analysis presented here illustrates a potential approach for the assessment of preventive health programs, emphasizing the importance of establishing clear objectives regarding cost and health outcomes over defined timeframes at the start of a vaccination program. The proposed success index offers a way to evaluate the performance of the program in the real world over time. Initially, the rotavirus vaccination campaign targeted what was perceived as a minor health concern in high-income countries. Disease elimination was anticipated through progressively expanding vaccine coverage, and some models predicted success when the vaccine was launched in 2006 [33,34]. Early vaccination outcomes exhibited notable reductions in rotavirus hospitalizations within the first two years, yet subsequent declines plateaued [26]. The absence of systematic, ongoing health gain monitoring meant that new, biennial disease peaks appearing some years after vaccine introduction, reflecting strategic waning of the vaccination, may not have always been reported. The RotaBIS study in Belgium addressed this lack of monitoring by initiating a routine annual data collection system once the vaccine was approved and reimbursed [28]. The study was able to maintain data processing for several years into the post-uptake period because it was collecting data recorded in routine practice and thus required a limited budget assignment. Finally, rigorous data analysis was crucial for identifying anomalous trends and understanding the underlying phenomena. The use of comparative analyses with subtly different scenarios provided valuable insights, informing interpretations of the observed data anomalies. This required the creation of two evaluation periods in the analysis, the use of two models, outcome measures assessed over two periods, and the development of a new impact calculation, familiar in the epidemiology world but relatively new in the health economics arena. The approach illustrated the necessity of understanding the data assembled, the effect of different infection sources on disease spread, and the effect on the vaccine impact of a sub-optimal introduction. The sensitivity analyses of worst-case and optimal vaccination launch scenarios were useful because different countries were found to be close to the results projected in both scenarios. The United Kingdom (UK) and Finland indicate what might be expected with a near-optimal rotavirus vaccine launch, compared with sub-optimal launches in Spain and Ireland [35,36,37,38]. The present analysis proposes an approach to investigating whether a vaccination program is successful using the success index ratio.

This comprehensive analysis illustrated the importance of considering impact evaluations in economic assessment, as was adopted in epidemiological research a few decades ago [8,39]. It could be argued that CEA will capture the impact from short to long durations in any case. However, it is important to make the distinction between the effects in populations at risk compared with the whole population. The effects in the wider population, not the population targeted by vaccination, are the cause of the new primary source of rotavirus infection (older children who in the pre-vaccination period had been a secondary source of infection) that appeared in the post-uptake period when the vaccination was not optimally introduced at the start. This new primary source will not be easy to target because the rotavirus vaccine does not directly reach the new source, as these children are too old to receive the vaccine. Any vaccination campaign introduced across a population may affect others besides those at-risk groups for whom vaccination is intended. Broader consequences may appear, which need to be captured in economic evaluations. It was an option to stop the RotaBIS monitoring study after eight years when the new infection equilibrium in the target population appeared to have been reached. Fortunately, the study continued to collect data over a longer period, which helped to construct a better explanation for the plateau reached during the uptake period, approximately three years after the vaccine’s introduction. The study’s emphasis on impact evaluations within the economic assessments of vaccines reflects a paradigm shift in health economics, akin to methodologies long employed in epidemiological research. This approach, exemplified by the rotavirus vaccination case study, acknowledges the broader societal impacts of vaccination beyond individual health gains. Sensitivity analyses exploring various vaccination scenarios further contextualized the findings and provided insights into optimal vaccination strategies.

The present analysis developed the success index because it may be useful for healthcare decisionmakers to have an unequivocal measure of the success of a vaccination program. The results observed in Belgium during the first two years after rotavirus vaccine introduction were considered a great success, as the country was the first high-income country to introduce the new vaccine systematically. However, our modeling analyses, supported by experience with rotavirus vaccine introduction in other countries, indicate that better outcomes could have been obtained if more detail about infection spread and disease burden had been available and applied to inform the implementation and timing of the vaccine’s introduction in Belgium. Our results may be useful for any new vaccine in current or future development [40].

## 5. Conclusions

Vaccines and infections need to be assessed individually to take into account their unique characteristics. Nevertheless, the assessment framework presented here has broad applicability. The evaluation approach, focusing on community infection control processes, transcends traditional cost-effectiveness analyses, offering a more comprehensive understanding of the value of a vaccination program expressed through its success index score from short- to long-term evaluation. Although developed with specific reference to the case of rotavirus vaccination, the principles and methodologies outlined in the present study are relevant for assessing the clinical and economic implications of future vaccine introductions using such a score.

## Figures and Tables

**Figure 1 vaccines-12-01265-f001:**
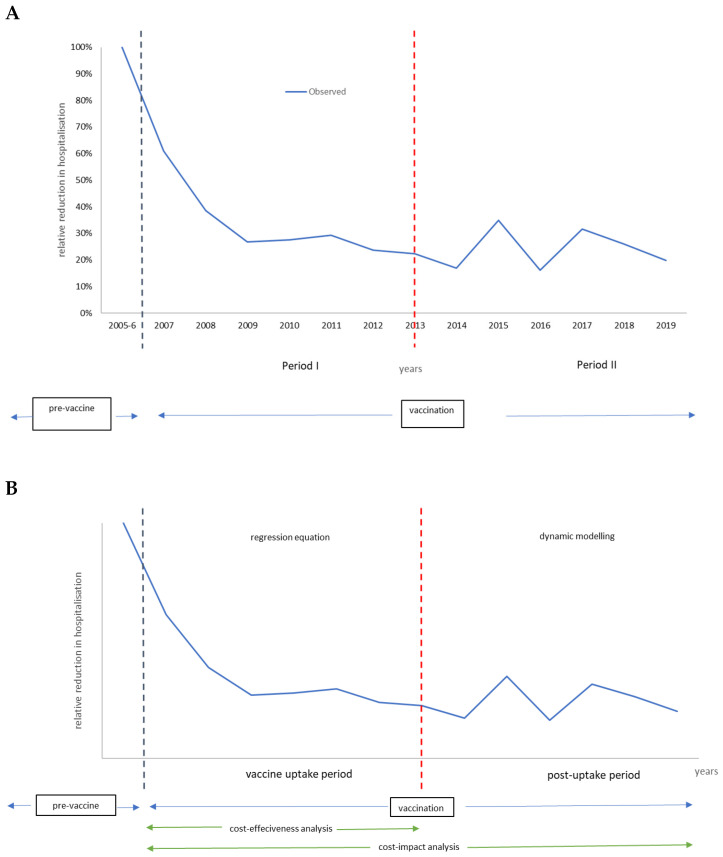
Defining two periods in the vaccination program model (**A**). The model specificities in each period and economic evaluation type (**B**).

**Figure 2 vaccines-12-01265-f002:**
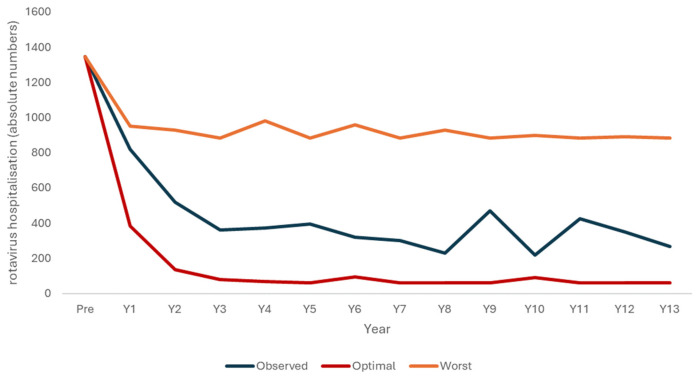
Three scenarios of rotavirus hospitalizations decreasing due to the vaccination program, with a modeled optimal launch scenario (red), the observed data (blue), and a modeled worst-case scenario (orange).

**Figure 3 vaccines-12-01265-f003:**
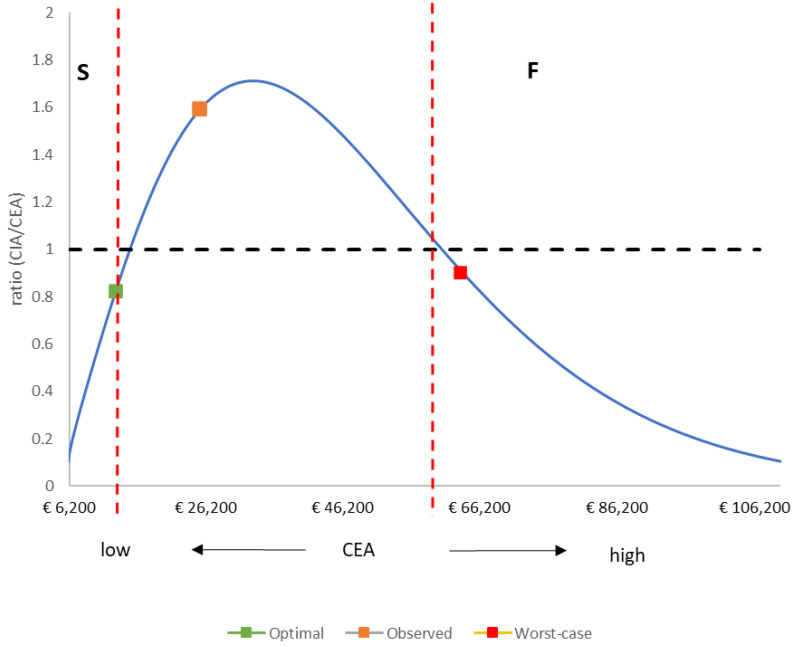
Outline of the success index ratio (CIA/CEA) as a function of the CEA value over an evaluation period of 15 years. CEA: cost-effectiveness analysis; CIA: cost impact analysis; S: Success; F: Failure.

**Table 1 vaccines-12-01265-t001:** Number of rotavirus hospitalizations by age and year (m—month; Yn—year number).

Age/Yn	2005–2006	2007	2008	2009	2010	2011	2012	2013	2014	2015	2016	2017	2018	2019
0–2 m	113	94	62	56	44	65	54	44	48	56	28	55	52	27
3–12 m	678	340	152	129	127	133	103	97	70	137	75	123	125	95
13–24 m	413	311	208	100	139	134	114	107	74	186	85	180	119	96
25–36 m	102	56	67	49	33	44	33	33	31	67	17	42	37	35
37–48 m	27	16	18	19	19	12	9	15	4	13	8	18	9	9
49–60 m	12	2	12	8	10	7	7	4	1	10	4	6	8	6
Total	1345	819	519	361	372	395	320	300	228	469	217	424	350	268

Colors were added to the different cells. They indicate who is under the influence of the vaccine (light green), + potential waning (dark green), primary herd effect (light and dark rose), secondary sources of infection (yellow), first vaccination (vaccine only (light blue), (+waning (dark blue)), secondary herd effect (grey), and new primary source (brown).

**Table 2 vaccines-12-01265-t002:** Cost and QALY loss input data used in the analysis.

Variable (Name)	Unit Value	Number	Total	Reference
Hospitalization pre-vaccination cost	€1467	7 days	€10,269	[14]
Hospitalization post-vaccination cost	€1467	5 days	€7335	[27]
Vaccine cost (Rotarix)	€70/dose	2	€140 /vaccination	[31]
QALY-loss pre	−0.47/hospital day	7 days	−0.009	[32]
QALY-loss post	−0.47/hospital day	5 days	−0.006	[27]
Target population to vaccinate pre-vaccination	5%	791	15,820	[11]

**Table 3 vaccines-12-01265-t003:** Key input data values for the uptake and the post-uptake period.

	Uptake Period	Variable Name	Post-Uptake Period
Vaccine efficacy	0.95	Average existing susceptible/wk	120
Vaccine coverage focused	0.66	Existing infectious/diseased/wk	1
Vaccine coverage routine	0.86	Birth rate increase/wk	20
Herd effect non-indicated	0.41	Force of infection	0.00833
Secondary infection source herd	0.10	Time unit (days)	3.5
Start month vaccination	Nov		

Focused: during the first months of vaccination before reaching the routine coverage; routine: reaching the normal coverage of child vaccination; herd effect non-indicated: herd effect amongst those who could not receive the vaccine; wk: week.

**Table 4 vaccines-12-01265-t004:** Cost-effectiveness results comparing days of hospitalization for no vaccination and vaccinated observed data for the vaccine uptake period.

Item	Age Group	No Vaccination	Vaccinated
Hospital days	0–2 m	904	467
	3–12 m	5424	1151
	13–24 m	3304	1187
	25–36 m	816	346
	37–48 m	216	112
	49–60 m	96	51
	Total	10,760	3314
Cost	Hospital cost	€15,784,920	€3,472,599
	Vaccine cost		€14,219,016
QALY	QALY-loss	−96.99	−21.34
	CEA		€25,204

m: month; QALY: quality-adjusted life years; €: Euro.

**Table 5 vaccines-12-01265-t005:** Cost-impact results comparing days of hospitalization regarding no vaccination, pre-launch predicted data, and the vaccinated observed data of the whole period.

Item	Age Group	No Vaccination	Vaccinated
Hospital days	0–2 m	1469	685
	3–12 m	8814	1706
	13–24 m	5369	1853
	25–36 m	1326	544
	37–48 m	351	169
	49–60 m	156	85
	Total	17,485	5042
Cost	Hospital cost	€25,650,495	€5,283,296
	Vaccine cost		€25,403,756
QALY	QALY-loss	−157.60	−32.46
	CIA		€40,247

m: month; QALY: quality-adjusted life years; €: Euro.

**Table 6 vaccines-12-01265-t006:** Ratio calculation of CEA and CIA for observed and simulated data (undiscounted).

	Difference in QALY-Loss	Difference in Cost	ICER	Ratio (CIA/CEA)
Observed				
Cost-effectiveness (CEA)	75.65	€1,906,695	€25,204	
Cost-impact (CIA)	125.14	€5,036,557	€40,247	1.59
Simulation Optimal launch scenario				
Cost-effectiveness (CEA)	55.94	€732,801	€12,939	
Cost-impact (CIA)	149.39	€1,599,297	€10,705	0.82
Simulation Worst-case launch scenario				
Cost-effectiveness (CEA)	26.58	€1,874,495	€70,507	
Cost-impact (CIA)	50.95	€3,224,655	€63,290	0.90

CEA, cost-effectiveness analysis; CIA, cost-impact analysis; ICER, incremental cost-effect ratio; QALY, quality-adjusted life-year.

## Data Availability

All of the data used are presented in previous publications. The analysis models developed are available upon request from the corresponding author.

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
