# Peer review of "Measuring the Vaccine Success Index: A Framework for Long-Term Economic Evaluation and Monitoring in the Case of Rotavirus Vaccination"

_vaccines, 2024, doi:10.3390/vaccines12111265_

Round 1

Reviewer 1 Report

Comments and Suggestions for Authors

The introduction of two rotavirus vaccines, RotaRix and RotaTaq, since 2006 has led to a significant reduction in rotavirus-related diseases. This study evaluated the success of rotavirus vaccination program in Belgium from an economic perspective, proposing a method to access the long-term health benefits and economic costs of rotavirus vaccination across the whole population. The authors analyzed cost-effectiveness of rotavirus in target population during the vaccine uptake period and the broader cost-impact spanning a longer period for both vaccinated and unvaccinated population. This article provides valuable information that would inform policymaking but could be improved with further edits.

Comments:

1.        Given that most children received their primary and booster vaccines prior to 32-week of age. It might take less than 6 years to cover the entire target population (infants and children under 5 years old). It would be helpful to explain why a 7-year period was selected as the "vaccine uptake period” in this study.

2.        In figure 2, the authors observed two hospitalization peaks in 9 and 11 years post-vaccine introduction. It would provide more insights, if the author could discuss possible factors contributing to these resurgences of rotavirus-caused hospitalization.

3.        In figure 3, specifying the labeling of x-axis would make it easier to interpret the data.

4.        Line 317-321: the authors mentioned that older children had been a secondary source of rotavirus infection. It would be helpful to clarify how older children contribute to transmission and include references to relevant studies.

5.        How is the disease-specific hospitalization defined in this study? Does it include only cases with acute gastroenteritis or laboratory-confirmed rotavirus infections are also required for categorization?

Author Response

The introduction of two rotavirus vaccines, RotaRix and RotaTaq, since 2006 has led to a significant reduction in rotavirus-related diseases. This study evaluated the success of rotavirus vaccination program in Belgium from an economic perspective, proposing a method to access the long-term health benefits and economic costs of rotavirus vaccination across the whole population. The authors analyzed cost-effectiveness of rotavirus in target population during the vaccine uptake period and the broader cost-impact spanning a longer period for both vaccinated and unvaccinated population. This article provides valuable information that would inform policymaking but could be improved with further edits.

Many thanks for taking time to review the manuscript.

Comments:

  1. Given that most children received their primary and booster vaccines prior to 32-week of age. It might take less than 6 years to cover the entire target population (infants and children under 5 years old). It would be helpful to explain why a 7-year period was selected as the "vaccine uptake period” in this study.

This is a very relevant point and it was also our first concern when analysing the observed data about when the uptake-period ends (period I) and when the post-uptake-period (period II) starts. As mentioned in the text we used two analysis models to replicate at best the observed data by splitting the whole observation period into two parts using for each part a different model type. The best match of both models together (regression equation for period I and an SIR model for period II) was for a situation when the first period ended at 7 years and the second period started at 8 years. We would have had more difficulties in well replicating the observed data through a modelling combination if period I ended at 5 or 6 years and period II started earlier. The point is that period I must have reached a new infection equilibrium situation in the target population before period II could start. The reach for the new infection equilibrium is locally conditional. It might take a longer period than the one we mathematically calculate as being 5 years for this child population and the vaccine effect, depending on the start of the vaccination program and the average annual vaccine coverage rate reached. We explored those points into previous publication as referred in the text: The economic value of rotavirus vaccination when optimally implemented in a high-income country. (1, 2)

  1. In figure 2, the authors observed two hospitalization peaks in 9 and 11 years post-vaccine introduction. It would provide more insights, if the author could discuss possible factors contributing to these resurgences of rotavirus-caused hospitalization.

We are making here reference to a publication that went more in depth on why those peaks may appear related to the start of the vaccination program and the presence/absence of primary and secondary sources of infection in the target group. As the paper here is not so much on the focus of what causes the appearance of new peaks but rather on the economic evaluation of long-term vaccination programs, we though that explaining the causes of the new peaks should be redundant to what has already been investigated and reported elsewhere. So, if people would like to better know, we like to refer to those previous publications (2).

  1. In figure 3, specifying the labeling of x-axis would make it easier to interpret the data.

We completely agree with the reviewer that the reported labelling of the X-axis is missing in Figure 3. Thank you for picking that up.

  1. Line 317-321: the authors mentioned that older children had been a secondary source of rotavirus infection. It would be helpful to clarify how older children contribute to transmission and include references to relevant studies.

That is a very good point that is not so easy to explain, unless we have the observed data reported as we did in Table 1. We have changed the colours in Table 1 to better explain the issue. Unfortunately, there is no other publication from another country that reported the data as we did to compare with. However, let explain in a nutshell how the process of hospital reduction is happening with the Belgian data as we have reported that point explicitly in 4 previous publications. I guess it is a little bit difficult to repeat that again in this paper because the focus here is more on how to report about the economic value on long-term vaccination and not about an explanation about how the observed data appear and why. During the vaccine uptake period there are 4 forces that impact the rate of hospitalisation linked to the start time of the vaccination and the vaccine coverage rate obtained at that start. The 4 forces that are present are the 2 direct vaccine related forces of vaccine effect and vaccine waning, and the 2 indirect vaccine forces related to the herd effect and the presence of secondary source of infection caused by the primary source of infection. Under ideal circumstances with a start of vaccination in June/July and a very high coverage at start (≥ 85%), first year subsequent the start of the vaccination you will have a big herd effect during that first year with no possibility of the development of secondary sources of infection because with an ideal start the primary source is fully attenuated that blocks the appearance of secondary sources. This good start is crucial for also having no secondary sources being present in the subsequent years. This has happened in the UK and in Finland (Finland just published a long term (10y) evaluation period that proofs the case(3)). If you don’t have that good start -like in Belgium we started in November instead of June/July- with no good vaccine coverage (≤ 85%), you then allow that secondary sources will manifest themselves by attenuating the herd effect in subsequent years. The clear and pure presence of the secondary source of infection is seen in one part of the table (yellow cells in Table 1) that is normally not impacted by the vaccine. In Finland and the UK, you will see low numbers, in Belgium you did not. That is the reason that secondary sources are present that will evolve over time in a new equilibrium of infection into a primary source… as seen in Belgium and reported by the SIR-model type. Hope this helps explaining what we see. This has been explicated in detail in the previous publications.(1, 2, 4-6)

  1. How is the disease-specific hospitalization defined in this study? Does it include only cases with acute gastroenteritis or laboratory-confirmed rotavirus infections are also required for categorization?

This has been reported in the previous publications on the protocol about how the data were collected. We thought that I would be redundant if we should start reexplaining the protocol again, but yes, a lab-test was essential to confirm rotavirus case hospitalisations.

  1. Standaert B. The economic value of rotavirus vaccination when optimally implemented in a high-income country. Vaccines (Basel). 2023;11:917.
  2. Standaert B, Benninghoff B. Defining the Recipe for an Optimal Rotavirus Vaccine Introduction in a High-Income Country in Europe. Viruses. 2022;14(2).
  3. Hemming-Harlo M, Gylling A, Herse F, Haavisto I, Nuutinen M, Pasternack M, et al. Long-term surveillance of rotavirus vaccination after implementation of a national immunization program in Finland (2008-2018). Vaccine. 2022;40(29):3942-7.
  4. Standaert B, Strens D, Raes M, Benninghoff B. Explaining the formation of a plateau in rotavirus vaccine impact on rotavirus hospitalisations in Belgium. Vaccine. 2022;40(13):1948-57.
  5. Raes M, Strens D, Vergison A, Verghote M, Standaert B. Reduction in pediatric rotavirus-related hospitalizations after universal rotavirus vaccination in Belgium. Pediatr Infect Dis J. 2011;30(7):e120-5.
  6. Standaert B, Strens D, Pereira P, Benninghoff B, Raes M. Lessons Learned from Long-Term Assessment of Rotavirus Vaccination in a High-Income Country: The Case of the Rotavirus Vaccine Belgium Impact Study (RotaBIS). Infect Dis Ther. 2020;9(4):967-80.

Reviewer 2 Report

Comments and Suggestions for Authors

Thank you for sharing your article on the long-term economic evaluation and monitoring of vaccine success by the rotavirus vaccine. The following minor comments may help when revising the manuscript:

L18/L20: Who should get reimbursed? Individual users or entire programs? Please revised the abstract accordingly.   

L99-104, L105-108: Lengthy sentences, please revise. 

L125: Please add a brief description of the hospitals mentioned to emphasise their representativeness.

Table 1: Please state in your manuscript how rotavirus infection was confirmed.

Figure 1: What is the time frame illustrated here for the vaccine update period and the post-uptake period? The figure should be self-explanatory.

L154: I may have missed this, but what is the time period of the post-uptake period? 

Table 2: Consider using the 1000 separator throughout for better readability. The same applies to Figure 4 and Table 5.

Author Response

Thank you for sharing your article on the long-term economic evaluation and monitoring of vaccine success by the rotavirus vaccine. The following minor comments may help when revising the manuscript:

L18/L20: Who should get reimbursed? Individual users or entire programs? Please revised the abstract accordingly.   

Thank you for making that point. The issue is effectively not about the reimbursement price given to the vaccine at start of the vaccination program, whether that price should be readjusted if the investment doesn’t lead to an optimal result. The issue is that the initiation of the vaccine programme was not done properly and that has led to no optimal success of the vaccine working. We have adjusted the text in the abstract.

L99-104, L105-108: Lengthy sentences, please revise. 

We did the revision. Thank you.

L125: Please add a brief description of the hospitals mentioned to emphasise their representativeness.

We add some description as requested. L125-126

Table 1: Please state in your manuscript how rotavirus infection was confirmed.

We add the lab-confirmation. L127

Figure 1: What is the time frame illustrated here for the vaccine update period and the post-uptake period? The figure should be self-explanatory.

We split the figure into 2 parts A and B which should help the reader to better explain the issue.

L154: I may have missed this, but what is the time period of the post-uptake period? 

In this analysis the post-uptake period is limited to 6 years from year 8 to year 13 post vaccine introduction. We could not continue the observation after 2019 because of the COVID infection that has dramatically impacted the rotavirus spread and I retired in 2020 and there was unfortunately no interest to continue the RotaBIS-study.

Table 2: Consider using the 1000 separator throughout for better readability. The same applies to Figure 4 and Table 5.

We adjust.